# The Use of Wound Infiltration for Postoperative Pain Management after Breast Cancer Surgery: A Randomized Clinical Study

**DOI:** 10.3390/biomedicines11041195

**Published:** 2023-04-17

**Authors:** Flaviu Ionut Faur, Ioana Adelina Clim, Amadeus Dobrescu, Alexandru Isaic, Catalin Prodan, Sabrina Florea, Cristi Tarta, Bogdan Totolici, Ciprian Duţă, Paul Pasca, Gabriel Lazar

**Affiliations:** 1IInd Surgery Clinic, Timisoara Emergency County Hospital, 300723 Timișoara, Romania; dr.faurflaviuionut@gmail.com (F.I.F.); isaicus@gmail.com (A.I.);; 2Department X of General Surgery, “Victor Babes” University of Medicine and Pharmacy, 300041 Timișoara, Romania; 3IInd Obstetric and Gynecology Clinic “Dominic Stanca”, 400124 Cluj-Napoca, Romania; 4Central Military Emergency University Hospital “Dr. Carol Davila”, 010825 Bucharest, Romania; 5Ist Clinic of General Surgery, Arad County Emergency Clinical Hospital, 310158 Arad, Romania; 6Department of General Surgery, Faculty of Medicine, “Vasile Goldiș” Western University of Arad, 310025 Arad, Romania; 7Department of Oncology Surgery, “Iuliu Hatieganu” University of Medicine and Pharmacy, 400347 Cluj-Napoca, Romania; 8Ist Clinic of Oncological Surgery, Oncological Institute “Prof Dr I Chiricuta” Cluj-Napoca, 400015 Cluj-Napoca, Romania

**Keywords:** postoperative pain, surgical procedure, local anesthesia

## Abstract

(1) Background: The present study aims to evaluate the reduction of postoperative pain in breast surgery using a series of local analgesics, which were infiltrated into the wound; (2) Methods: Envelopes containing allocation were prepared prior to the study. The patients involved were randomly assigned to the groups of local anesthesia infiltration (Group A) or normal pain management with intravenous analgesics (Group B). The random allocation sequence was generated using computer-generated random numbers. The normally distributed continuous data were expressed as the means (SD) and were assessed using the analysis of variance (ANOVA), independent-sample *t*-test, or paired *t*-test; (3) Results: The development of the postoperative pain stages was recorded using the VAS score. Therefore, for Group A, the following results were obtained: the VAS at 6 h postoperatively showed an average value of 0.63 and a maximum value of 3. The results for Group B were the following: the VAS score at 6 h postoperatively showed an average value of 4.92, a maximum of 8, and a minimum of 2; (4) Conclusions: We can confirm that there are favorable statistical indicators regarding the postoperative pain management process during the first 24–38 h after a surgical intervention for breast cancer using local infiltration of anesthetics.

## 1. Summary

Breast cancer is amongst the most common types of cancer affecting women worldwide, with an improved diagnosis and management leading to a 5-year survival rate in over 80% of cases [1,2,3,4,5,6,7,8]. Moreover, regular preoperative, intraoperative, and postoperative check-ups of breast cancer represent a very important aspect in the optimization and improvement of the results in the short, medium, and long term [9,10,11]. In reality, the management of postoperative pain is the real interest throughout the post-surgery period

In the literature, the opinions on postoperative pain management in the case of oncological interventions for breast cancer are quite uniform, with no standard gold procedure established [12,13].

Opioids are commonly used for perioperative analgesia in breast cancer surgery. However, their use is associated with several potential side effects, including respiratory depression, nausea, vomiting, sedation, and constipation.

There are several important factors to consider when using opioids in perioperative analgesia for breast cancer: (1) Individualization of treatment: The dose and duration of opioid therapy should be individualized based on the patient’s medical history, pain severity, and the type of surgery performed. Patients with a history of substance abuse or addiction may require alternative pain management strategies; (2) Multimodal analgesia: Opioids should be used as part of a multimodal analgesic approach that includes non-opioid medications, such as acetaminophen, non-steroidal anti-inflammatory drugs (NSAIDs), and local anesthetics. This approach can reduce opioid use and minimize opioid-related side effects; (3) Monitoring for side effects: Patients receiving opioids for perioperative analgesia should be closely monitored for side effects, including respiratory depression, sedation, nausea, and vomiting. Patients should also receive prophylactic treatment for opioid-induced constipation; (4) Opioid-sparing techniques: Opioid-sparing techniques, such as nerve blocks and epidural analgesia or a wound infiltration technique, can be used to reduce opioid use and minimize side effects. These techniques may be particularly useful for patients undergoing mastectomy or breast reconstruction.

This last component of the opioid-sparing technique leads to the central theme of this study, specifically, Tumescent Local Anesthesia (TLA), which was first described in 1987 by Klein (1987). This technique was used to provide analgesia for liposuction without general anesthesia and postoperative analgesia in human patients (Klein 1988) [14,15]. The procedure has been used for mastectomy surgery as adjunct general anesthesia in veterinary medicine [16]. Using local infiltrative anesthesia, where the drug is injected directly into the subcutaneous tissue or surgical wound at the end of the surgery, is a safe and effective way to improve postoperative pain and the recovery of the functional status [17,18]. From a pharmacological point of view, these analgesics have a wide range of mechanisms of action, such as modulating voltage-gated channels, ligand-gated channels, receptors, and cellular pathways [19,20]. Examples of the advantages include performing unique special analgesia intraoperatively, limiting the systemic side effects of conventional analgesia, and reducing prices through postoperative analgesia.

## 2. Data Description

The study took place at the First Clinic of Oncological Surgery, The Oncology Institute “Prof Dr I Chiricuta”, Cluj-Napoca, Romania between January 2021 and July 2022, having a prospective and randomized nature.

It included 76 patients who went through a series of oncological surgical interventions around the breast area (breast-conserving surgery–BCS, oncoplastic breast conservation surgery–OCBS, modified radical mastectomy–MRM).

For the good management of the postoperative local pain process, we considered it necessary to include an individualized procedure that consisted of performing a locoregional infiltration at the excisional site level using a mix of substances meant to temper the pain, both in the short and long term. The individualized procedure requires mixing 10 mL Lidocaine 1% plus 10 mL Ropicavaina 1% and for the cytokinin blockade, 1 mL Ketorol plus 2 mL Dexamethasone. The patients were randomly allocated into two groups (Group A-38 patients-Locoregional wound infiltration and Group B-38 patients-Normal postoperative pain management) on a 1:1 ratio by drawing blocks of four patients so that, in the event of the suspension or discontinuation of the study, the number of patients would remain similar in both groups. We also used a database where we recorded several analytical standard criteria that were particular for each group (BMI—Body Mass Index, ASA score-American Society of Anesthesiologists physical status classification system, molecular subtype according to the St Gallen classification, tumor histology, surgery type, VAS at 6, 12, 24, and 36 h postoperatively, amount of drain fluid at 12, 24, and 48 h postoperatively, time of drainage suppression, quantity of painkillers administrated postoperatively, number of days of hospitalization).

## 3. Methods

Prior to the initiation of the study, 78 sequentially numbered envelopes containing the allocation were prepared. The patients involved were randomly assigned to the groups of local anesthesia infiltration (Group A) or normal pain management with intravenous analgesics (Group B). The random allocation sequence was generated using computer-generated random numbers. SPSS 23.0 for Windows (SPSS, Inc., Chicago, IL, USA) was used for data analysis. The normally distributed continuous data were expressed as the means (SD) and were assessed using the analysis of variance (ANOVA), independent-sample t-test, or paired *t*-test. The nonparametric data were analyzed using the Mann–Whitney and Wilcoxon tests. Two-sided tests were performed to declare the statistical significance at *p* < 0.05. Ethical clearance was obtained from the institutional ethics committee (The Ethics Commission for Research and Development Activities and the Quality Assurance of Clinical Trials of the “Prof. Dr. Ion Chiricuță” Oncology Institute in Cluj-Napoca, appointed by decision of the manager (IOCN nr. 189 -03.06.2021, Application nr. 10442)) according to the rules of the Declaration of Helsinki of 1975.

## 4. Personal Technique

For the anesthetic technique, general anesthesia using Sevoflurane was opted for. For the anesthetic induction, the following substances were used: 0.07–0.1 mg/kg iv Midazolam, 2–20 µg/kg Fentanyl, 1.5–2.5 mg/kg Propofol, 0.6 mg/kg Rocuronium. On the other hand, in order to protect the airway, an endotracheal tube was used, followed by mechanical ventilation under Sevoflurane depending on adjusted parameters. During the postinduction phase, 4–8 mg of Dexamethasone was administrated for the antiemetic and anti-inflammatory effect with the aim of boosting the local anesthesia effect. For surgical stress ulcer prophylaxis, we used 40 mg of Proton Pump Inhibitors (PPI). To reverse the neuromuscular blockade, we used 2–4 mg/kg of Sugammadex. To prevent postoperative dyspeptic syndrome (nausea, vomiting), we administrated 8 mg of Ondansetron 30–40 min before extubation.

The standard intravenous analgesia applied postoperatively to the group of patients who did not experience locoregional wound infiltration (Group B) was performed using non-opioid medication: 500–1000 mg Paracetamol every 6 h, 500 mg/2 mg/0.02 mg/mL Algifen every 8 h, 20 mg Nefopam every 8 h, and 50 mg Dexketoprofen every 12 h.

To perform an examination of the locoregional pain during surgery on the mammary gland, we decided to carry out a locoregional anesthetic block (Group A) using 1% Lidocaine–10 mL (short-acting local anesthetic) and 1% Ropivacaine–10 mL (long-acting local anesthetic), respectively. To create the local cytokine blockade, we used a steroidal anti-inflammatory agent–1 mL/30 mg Ketorol and a non-steroidal anti-inflammatory agent–2 mL/8 mg Dexamethasone, respectively. The injection was performed systematically after excision, both at the base of the dermo-glandular/dermo-adipose flap and at the incision-line level. The injection was performed using a syringe of 20 mL to which a 23G needle was attached.

## 5. Primary Outcomes

The pain was immediately assessed after returning to the post-anesthesia care unit (PACU) and 6 h, 12 h, 24 h, and 36 h after surgery using a VAS score (0 = no pain to 10 = most severe pain). The dynamic analysis of the VAS was performed comparatively between the two groups of study.

## 6. Secondary Outcomes

A secondary outcome was that of the analysis of the amount of drain fluid 12, 24, and 48 h postoperatively, the moment of drainage suppression, the quantity of painkillers administrated postoperatively, the number of days spent in the hospital depending on the type of surgical intervention, and the VAS dynamics. The analysis was made in a comparative manner. Postoperative drainage analysis is important because it provides information about the level of fluid accumulated locally and the possibility of a collection, having also a therapeutic role in its drainage.

## 7. Patient Autonomy/Ethical Consent

All the procedures approached throughout the entire study that involved human subjects were approved by the Ethics Commission according to the national and international standards directly related to the declaration of Helsinki in 1964. This article does not include any study on lab animals. The consent mentioned above was received from and approved by each participant in the study (The Ethics Commission for Research and Development Activities and for the Quality Assurance of Clinical Trials of the “Prof. Dr. Ion Chiricuță” Oncology Institute in Cluj-Napoca, appointed by decision of the manager (IOCN nr. 189 -03.06.2021, Application nr. 10442)).

## 8. Results

In this prospective and randomized study, a total of 76 patients underwent a series of oncological surgical interventions around the breast area. The patients were randomly divided into two groups, i.e., Group A and Group B, with the aim of evaluating various analytical parameters (BMI, ASA score, the molecular subtype according to the St Gallen classification, the tumor histology, the surgery type, the postoperative VAS at 6, 12, 24, and 36 h, the amount of drain fluid at 12, 24, and 48 h postoperatively, the time of drainage suppression, the quantity of pain killers administrated postoperatively, the days of hospitalization) and testing the feasibility of creating a locoregional anesthetic block intended to alleviate postoperative pain. As previously mentioned, the general batch of patients was divided into two groups: Group A (38 patients) who were injected locoregionally with a combination of substances (1% Lidocaine, 1% Ropivacaine, 30 mg Ketorol, 8 mg Dexamethasone) and Group B (38 patients) who were administrated a standard intravenous painkiller treatment.

According to the St Gallen Classification (Table 1), and with reference to the general batch, the following values were recorded: 31.57% of the cases (*n* = 24) were categorized as Luminal A, 22.36% (*n* = 17) were TNBC (Triple-negative breast cancer) type (triple negative cancer), and 19.73% were Her2+, while 14.47% (*n* = 11) were categorized as Luminal B Her2 and 11.84% (*n* = 9) were Luminal B Her2+. Moreover, from the point of view of the histological type of the tumor at the general batch level (Table 1), the majority of the cases presented an infiltrative ductal carcinoma (82.89%, *n* = 63), followed by a lobular pattern in 9.21% of the cases (*n* = 7) and smaller percentages in the case of metaplastic (5.26%) and medullary carcinoma (2.63%).

From the point of view of neoadjuvant therapy (NAC), 44.73% of the patients (*n* = 34) had NAC as their primary therapeutic step, while 55.26% had surgery as their primary therapeutic step. The case distribution based on the TNM classification was as follows: Stage I–18.42%, stage II–51.31%, stage III–21.05%, and stage IV–9.21%.

### 8.1. Group A

Group A included 38 patients aged between 31 and 76 years, with the mean age being 50.34 ± 3.4 years. These patients received locoregional anesthesia and dermo-glandular/dermo-adipose infiltration (the technique described above) at the excision level to moderate the postoperative painful period. The patients belonging to Group A had an average ASA score of 2.15, the minimum was 1 and the maximum was 3. The average body mass index (BMI) of the group was 30 kg/m^2^; the minimum was 23 kg/m^2^ and the maximum was 36 kg/m^2^. As for the postoperative drainage suppression, the average time was 5.86 ± 2.3 days; the minimum was 4 days and the maximum was 9 days. The average number of days of hospitalization was 2.36 ± 1.2, with a minimum of 2 days and a maximum of 4 days. With regard to the quantity of painkillers administrated additionally after surgery, the average was 0.57 vials per patient, and the maximum value was 2 vials per patient. On the other hand, most of the patients presented few painful stages after locoregional injection. This eliminated the necessity of administrating any additional painkillers (*p* > 0.001). The patients who did not have significant postoperative pain did not receive additional analgesics.

By analyzing the data from Table 2, we can notice the way in which the cases were distributed based on the type of surgical intervention. In Group A, 68.41% undertook breast-conserving surgery (BCS) in 44.73% of the cases, while 23.68% of the cases had oncoplastic breast-conserving surgery (OBCS). There is another percentage of 31.57% of the cases where the patients had a radical modified mastectomy (Madden procedure) without per primam reconstruction (Table 3).

The development of the postoperative pain stages was recorded using the VAS score. Therefore, for Group A, the following results were obtained: the VAS at 6 h postoperatively showed an average value of 0.63 and a maximum value of 3; the VAS at 12 h postoperatively showed an average value of 0.84 and a maximum value of 3. The analysis of the VAS at 24 h postoperatively showed an average of 0.71 and a maximum value of 2, while the VAS at 36 h showed an average of 1.1 and a maximum value of 3. By examining the data obtained, an efficacity of the locoregional infiltration for pain management during the first 24 h after surgery (*p* < 0.001) can be observed.

When it comes to the dynamics of the amount of drain fluid, it can be observed that there is an efferent peak at 24 h postoperatively, with an average of 93/68 mL, a minimum of 60 mL, and a maximum of 130 mL. Hereafter, the amount of the output starts decreasing (Table 4).

By analyzing Figure 1, it can be observed that there is a logarithmic decrease in the VAS during the first 36 h post-surgery, showing undervalue points.

By analyzing Figure 2, the logarithmic curve of the dynamic development of the amount of the drain fluid can be observed, with maximum values observed during the first 24 h post-surgery, followed by a linear phase, then a descended phase at 48 h postoperatively.

### 8.2. Group B

Group B included 38 patients aged between 30 and 75 years, the mean age being 49.55 ± 2.4 years. For their postoperative pain management process, they were administered analgesics and anti-inflammatory medication (steroidal and non-steroidal) at fixed timeframes based on a standardized protocol. In terms of the general characteristics of Group B, the patients presented an average ASA score of 2.10, with a minimum of 1 and a maximum of 3. The average body mass index (BMI) was 29.44 kg/m^2^, with a lowest value of 23 kg/m^2^ and a highest value of 35 kg/m^2^. For the postoperative drainage suppression, the average time was 10.10 ± 4.7 days, with a minimum of 7 days and a maximum of 15 days. The average number of days the patients were hospitalized was 2.86 ± 1.2 days, with a minimum of 2 days and a maximum of 4 days. If we are to refer to the quantity of analgesics and anti-inflammatory medication administrated additionally post-surgery, the average was 13.50 vials per patient, the minimum was 9 vials per patient, and the maximum was 17 vials per patient. The majority of the subjects in Group B reported pain during the first 36 h post-surgery, with an average VAS score between 2.2 and 4.9. Therefore, the administration of analgesics was necessary based on the standardized protocol mentioned above.

The data presented in Table 5 highlight a classification of the cases in Group B based on the type of surgical intervention, as follows: 63.15% of the cases undertook breast-conserving surgery (BCS in 36.84% of the cases and OBCS in 26.31% of the cases), whereas 34.21% of the cases had a Radical Modified Mastectomy (Madden procedure) type of surgery without per primam reconstruction.

From the point of view of the type of mammoplasty, 10.52% of the cases had the Riberio–Robbins type, 7.89% had the Grisotti type, 5.26% had the Wise type, and 2.63% had the Pitangui–Lejour type (Table 6).

Similar to Group A, the development of the pain stages post-surgery was evaluated using the VAS score, and the results for Group B were the following: the VAS score at 6 h postoperatively showed an average value of 4.92, a maximum of 8, and a minimum of 2; the VAS score at 12 h postoperatively showed an average value of 3.71, a maximum of 6, and a minimum of 2; the VAS score at 24 h postoperatively showed an average value of 2.71, with a maximum of 4; the VAS at 36 h post-surgery showed an average of 2.28 and a maximum of 3. Based on these data, it can be noticed that the peak was somewhere between 12 and 24 h post-surgery. After that, the pain level reached a linear phase, which was conserved up to 36 h postoperatively.

Concerning the dynamics of the drain fluid volume, it can be noticed that the peak was somewhere at 12 h post-surgery, with an average of 128.42 mL, a minimum of 70 mL, and a maximum of 210 mL, while later on it reached a linear phase (Table 7).

The data presented in Figure 3 highlight a logarithmic decrease in the VAS during the first 36 h postoperatively, showing undervalue points.

The data shown in Figure 4 highlight a logarithmic curve of the dynamic development of the amount of drain fluid with maximum values during the first 24 h post-surgery, followed by a linear phase and then a decreasing phase at 48 h postoperatively.

By making a comparison between the data obtained for Group A and the data obtained for Group B, it can be noticed that the VAS dynamics in Group A were significantly reduced (*p* < 0.001). The same applied to the drain fluid (*p* < 0.005), which highlights a more effective result of the locoregional infiltrative anesthesia and, at the same time, a better effect of the cytokinin blockade on the local exudative process. As previously presented, by analyzing the VAS score 6 h after surgery, we obtained an average of 0.63 for Group A vs. 4.92 for Group B (*p* < 0.001) and an average VAS score of 0.84 vs. 3.71 (*p* < 0.001) 12 h post-surgery. Similarly, 24 h after surgery, we obtained an average VAS score of 0.71 for Group A vs. 2.71 (*p* < 0.001) for Group B, while at 48 h after surgery, the values were 1.1 for Group A and 2.28 (*p* < 0.005) for Group B. By assessing the VAS data obtained for the two groups, we managed to obtain statistical coefficients regarding post-surgery pain management.

By analyzing the dynamics of the drain fluid, a decrease in the drain fluid volume can also be observed for Group A, whose subjects received a locoregional cytokinin blockade compared to Group B, whose subjects did not. Therefore, the average amount at 12 h for Group A was 44.73 mL vs. 83.16 mL for Group B (*p* < 0.002); the average amount at 24 h was 93.68 mL for Group A vs. 128.42 mL (*p* < 0.002) for Group B; and the average at 48 h was 78.68 mL for Group A vs. 124.74 mL (*p* < 0.005) for Group B. In this way, we managed to show that we had a significant decrease in the liquid dynamics in the case of the group that received locoregional infiltration based on anesthesia and analgesia as additional procedures.

According to Figure 5, for drainage suppression, the average time was 5.86 ± 2.3 days, with a minimum of 4 days and a maximum of 9 days for Group A vs. 10.10 ± 4.7 days, a minimum of 7 days, and a maximum of 15 days for Group B. By performing a retrospective analysis of the Group A results, we can highlight a limitation of the amount of drain fluid after the first 24 h, followed by a logarithmic decrease at 48 h postoperatively.

With reference to the quantity of painkillers additionally administrated (Table 8), the average value was 0.57 vials per patient, with a maximum value of 2 vials for Group A vs. an average of 13.6 vials per patient and a maximum of 17 vials for Group B.

## 9. Discussion

In this section, multiple previous studies are described and a number of comparisons between these studies and what is happening worldwide are made.

In 2008, Baundray et al. promoted a study in which they analyzed the efficiency of local wound infiltration using Ropivacaine post-MRM and sectorectomy with axillary lymph node dissection (ALND), a study in which they obtained a set of statistically insignificant parameters when comparing the study groups in terms of postoperative pain management [21,22,23,24,25,26,27]. Two other studies published in 2000 and 2003 by Johansson et al., which also focused on analyzing the efficiency of post-surgery pain management through local wound infiltration using Ropivacaine for patients who had sectorectomy with or without ALND, obtained similar results that were statistically insignificant in favor of wound infiltration [28,29,30,31,32]. In 2004, Talbot et al. published a study in which they analyzed the efficiency of local instillation for postoperative pain management for patients who had an MRM, using Bupivacaine 0.5% mixed with 20 mL saline solution [33]. The procedure applied by the authors throughout the study was that of the instillation of the analgesic substance in the underarm area using drain tubes. The results of this study did not present any statistically significant values of the analyzed parameters.

Another study published in 2011 by Vigneau et al. tried to show the efficiency of wound injection (post-MRM and ALND sectorectomy) with 7.5 mg/mL Ropivacaine mixed with 20 mL saline solution. The authors obtained a set of statistically significant parameters regarding the pain management process 2, 4, and 6 h postoperatively (study group vs. control group) [34].

In 2013, Albi-Feldzer et al. published a study in which they evaluated and analyzed postoperative pain management for a group of patients who had MRM and BCS with or without ALND, for whom they used Ropivacaine 0.375% mixed with saline solution. The infiltration was performed at the intercostal level in spaces 2 and 3 and in the area of the humeral insertion of the pectoralis major muscle. The results of the Albi-Feldzer et al. study showed statistically significant values regarding postoperative pain management for the study group vs. the control group. Another study analyzed by us was the one published by Nirmal et al. in 2019 who used Ropivacaine–0.25% mixed with 40 mL of saline solution for wound infiltration post-MRM. The authors obtained significant results for post-surgery pain management at 15 h compared to the control group [35,36].

## 10. Conclusions

To conclude, we can confirm that there are favorable statistical indicators regarding the postoperative pain management process during the first 24–38 h after a surgical intervention for breast cancer using the local infiltration of anesthetics and, at the same time, by reducing the drain output with the help of a local cytokinin blockade, the reduction of painkillers, and the minimization of serous fluids. This analgesia technique can be envisaged for enhancing recovery after surgery in both hospital and outpatient settings. Further studies should be performed to confirm this possibility and introduce a gold-standard technique.

## Figures and Tables

**Figure 1 biomedicines-11-01195-f001:**
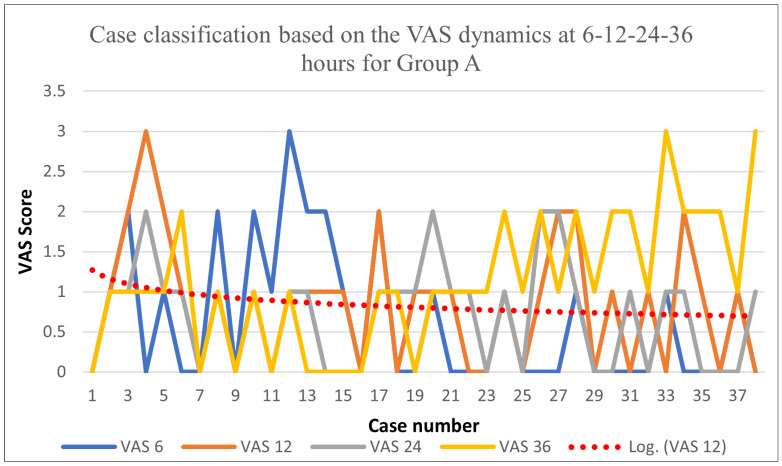
VAS dynamics for Group A.

**Figure 2 biomedicines-11-01195-f002:**
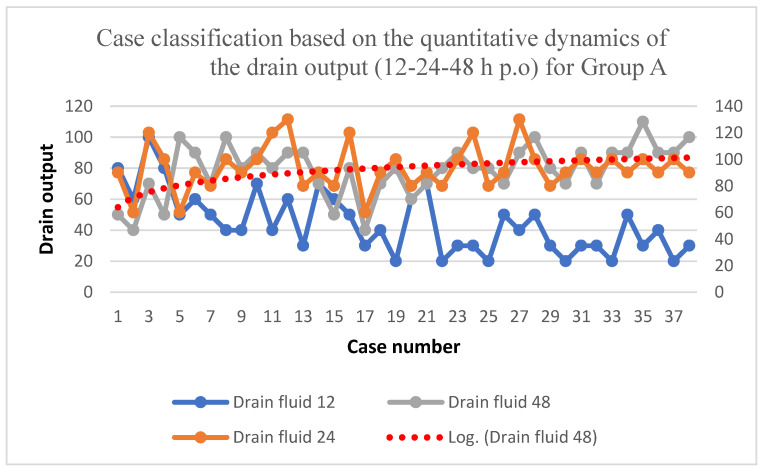
Quantitative dynamics of the drain output during the first 48 h for Group A.

**Figure 3 biomedicines-11-01195-f003:**
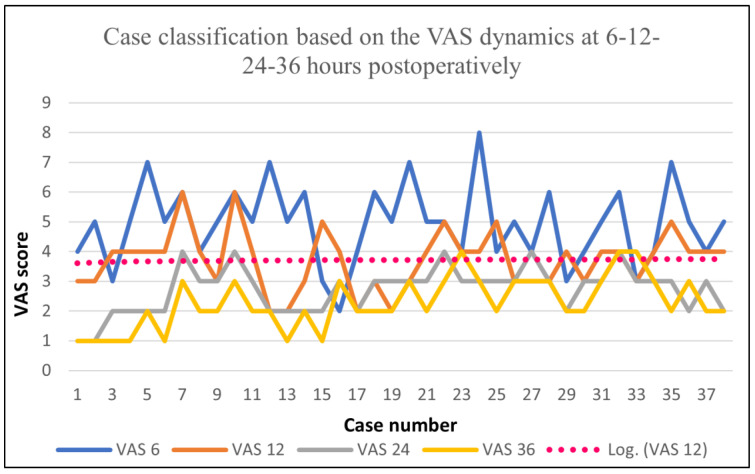
VAS dynamics for Group B.

**Figure 4 biomedicines-11-01195-f004:**
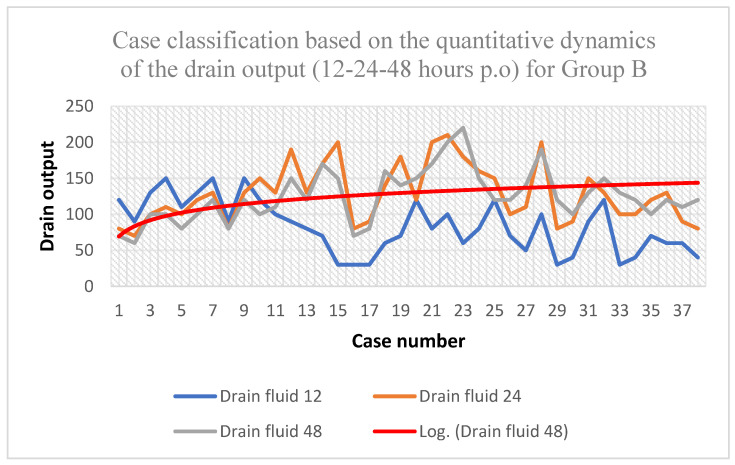
Quantitative dynamics of the drain output during the first 48 h for Group B.

**Figure 5 biomedicines-11-01195-f005:**
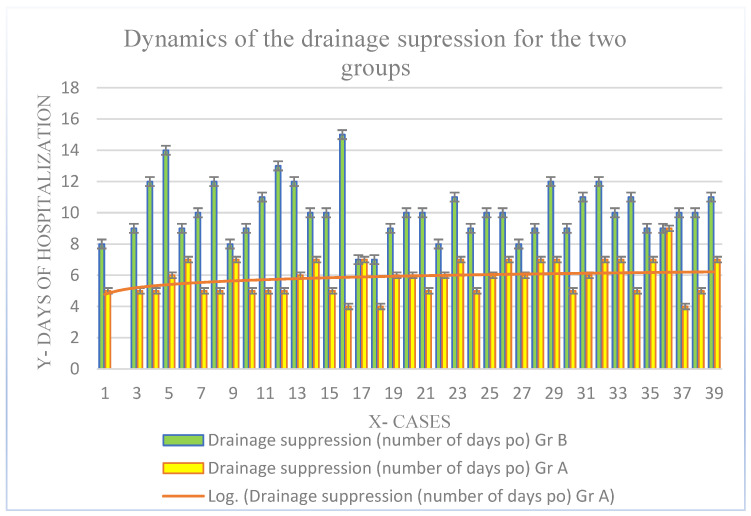
Time required for drainage suppression. A comparison between the two groups.

**Table 1 biomedicines-11-01195-t001:** Patient classification based on the molecular subtype, histological type, and presence/absence of NAC.

Parameters	No. of Cases	%
Luminal A	24	31.57
Luminal B Her2 -	11	14.47
Luminal B Her2 +	9	11.84
Her2+	15	19.73
TNBC	17	22.36
Histologic type		
Infiltrative ductal carcinoma	63	82.89
Lobular	7	9.21
Metaplastic	4	5.26
Medullary	2	2.63
NAC therapy		
NAC	34	44.73
Non-NAC	42	55.26

**Table 2 biomedicines-11-01195-t002:** Case classification based on surgery type for Group A.

Type of Intervention		%
BCS	44.73%	
Total % of BCS + OBCS		68.41%
OBCS	23.68%	
MRM (Madden procedure)	31.57%	31.57%

**Table 3 biomedicines-11-01195-t003:** Case classification based on the type of mammoplasty performed in Group A.

Mammoplasty	No. of Cases	%
Pitangui–Lejour	4	10.52
Wise pattern	2	5.26
Riberio–Robbins	3	7.89
Total	9	23.68

**Table 4 biomedicines-11-01195-t004:** An analysis of the VAS dynamics and amount of drain fluid for Group A.

Function	VAS 6	VAS 12	VAS 24	VAS 36
Average	0.63	0.84	0.71	1.1
Min	0	0	0	0
Max	3	3	2	3
	Drain fluid 12 (mL)	Drain fluid 24 (mL)	Drain fluid 48 (mL)	
Average	44.73	93.68	78.68	
Min	20	60	40	
Max	100	130	110	

**Table 5 biomedicines-11-01195-t005:** Case classification based on surgery type for Group B.

Type of Intervention	%	Total %
BCS	36.84%	
		63.15
OBCS	26.31%	
MRM (Madden procedure)	34.21%	34.21

**Table 6 biomedicines-11-01195-t006:** Case classification based on the type of mammoplasty performed in Group B.

Mammoplasty Type	Case Number	%
Pitangui–Lejour	1	2.63
Wise pattern	2	5.26
Riberio–Robbins	4	10.52
Grisotti	3	7.89
Total	10	26.31

**Table 7 biomedicines-11-01195-t007:** An analysis of the VAS dynamics and amount of drain fluid for Group B.

Function	VAS 6	VAS 12	VAS 24	VAS 36
Average	4.92	3.71	2.71	2.28
Min	2	2	1	1
Max	8	6	4	4
	Output 12 (mL)	Output 24 (mL)	Output 48 (mL)	
Average	83.16	128.42	124.74	
Min	30	70	60	
Max	150	210	220	

**Table 8 biomedicines-11-01195-t008:** Paramedian and comparative distribution at the general batch level.

Type of Surgery	Group A	Group B	*p* Value
BCS	44.73%	36.84%	
OBCS	23.68%	26.31%	0.965
RMM	31.57%	34.21%	
Surgical drain fluid volume (mean value in mL at 12–24–48 p.o)
12 h p.o	44.73 mL	83.16	
24 h p.o	93.68 mL	128.42	0.001
48 h p.o	78.68	124.74	
Complications			
Haematoma	1	2	
Seroma	1	4	0.05
Days of hospitalization (mean value ± SD)	2.36 ± 1.2	2.86 ± 1.2	0.67
The number of analgesics administered p.o (mean value)	
Average	0.57	13	
Min	0	9	0.001
Max	2	17	
Smoking	42.10% (*n* = 16)	36.84% (*n* = 14)	0.889
No smoking	57.89% (*n* = 22)	63.15 (*n* = 24)

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
