# Peer review of "The Use of Wound Infiltration for Postoperative Pain Management after Breast Cancer Surgery: A Randomized Clinical Study"

_biomedicines, 2023, doi:10.3390/biomedicines11041195_

Round 1
Reviewer 1 Report
The paper describes the clinical results of a clinical study about the pain management in patients affected by breast cancer. The authors demonstrate that they have not paid sufficient attention to how the text should be structured in compliance with the rules indicated by the journal. Moreover, the work is not very clear and is poorly exposed. The introduction is very sparse and little emphasis is given to the purpose of the work. As indicated in the instructions, the figures and tables must always be cited in the text and added just after having mentioned them, for a better understanding by the reader. Overall, the work needs a restructuring and a deepening in the introductory part and in the discussion.
Specific comments
· The Authors should reread carefully the Instructions for Authors to prepare their manuscript. The abstract should be a total of about 200 words maximum, not fractioned in subtitles, no one references must be reported
· Line 65, the Country should be added.
· Line 73 It is not clear the amount of drug administered. 1ml/30mg of Ketorol, 2ml/8mg of Dexamethasone, what do they mean?: Can the Authors provide the all the concentrations of the drugs administered in a more proper way? For example, mg of drug per volume of injected solution.
· The Introduction is very poor. The Authors should explain the reasons underlying their research. Why did they consider not satisfactory the standard intravenous painkiller treatment? How were selected alternative drugs? Are they employed in other circumstances or protocols? Their selection should be supported and explained by references.
· Table 1 must be cited in the text. Add a footnote for the explanation of the acronyms NAC and TNBC.
· Titles for x and y axis are necessary in the figures.
· Figure 5 and table 8 must be cited in the text.
· Please use L as unit of measurement of liter, so mL and not ml.
· Table 8 check the numbers, some have the coma.
· The discussion seems an introduction and should be transferred to this section. In the discussion should be highlighted the improvement of the approaches tested in the study or at least a comparison should be made
Author Response
Greetings!
I managed to complete the changes requested by you in the review report!
Please see the attachment!
Have a nice day!

Reviewer 2 Report
see attached

Author Response
Greetings!
We have made the changes specified by you in the report, these being marked in green in the attached document.
Please see the attachment
Have a nice day!

Reviewer 3 Report
This article tends to evaluate the benefits of a non-traditional pain management that uses a mix of anesthetic and anti-inflammatory reagents to perform locoregional injection for wound infiltration of these compounds to temper the pain of postoperative breast cancer patients. The authors compared such pain management with the standard intravenous painkiller treatment. They found favorable statistic indicators regarding the local wound infiltration postoperative pain management for breast cancer patients. Moreover, painkillers and serous fluids (drainage output) were reduced due to a local cytokines blockade compared to those who used traditional pain management. They hope this type of pain management shall become a gold standard technique in the future after further studies and optimization. Overall, the manuscript was well-written and easily understandable. The biological significance and clinical impact are relatively high. However, there are some suggestions to make the manuscript more understandable.
1. How were the dosages of the mix of substances used to perform a locoregional infiltration at the excisional site determined? Could the ingredients and doses in the mixture be changed and optimized to best benefit breast cancer patients?
2. The authors should give the titles of the Y axes in Fig. 1 to 4 and explain what the numbers mean.
3. The text, Tables, and Figures belonging to Group A and B should be combined and not separated so that the comparisons between both groups are much easier for readers.
4. The references should be updated to 2023 if possible.
Author Response
We have made the changes specified by you in the report, these being marked with green and yellow in the attached document.
We managed to improve the terms using the specified linguistic requirements.
Please see the attachment

Round 2
Reviewer 1 Report
The Authors did not provide neither a file with point by point answers, nor a manuscript with highlighted the changes, so I'm forced to reject the paper
Author Response
I have made the changes requested by you
Thanks for the guidance and support

Reviewer 2 Report
The manuscript is improved and is ready for publication.
Author Response
We managed to fulfill your requirements regarding the changes that had to be made within the article. Thank you for your support and patience

Reviewer 3 Report
The way the authors addressed reviewers' comments was not very professional and not acceptable. They should separately respond to each reviewer and address the comments point-by-point.
After reading the authors' responses and the revised version, I'm not satisfied with the changes in response to my previous comments #1, 3, and 4. Without sincerely and adequately addressing these comments, the current form of the manuscript is not recommended to be published in Biomedicines.
Author Response

(The authors gave the same response as above.)

Round 3
Reviewer 1 Report
Dear Authors,
I assume you know how responses to reviewers are written. the attached file contains generic responses to all reviewers at once. therefore, I confirm the decision already made which is to reject the work. you have to provide an individual letter with the lines marked, and a text with the changes made highlighted
Author Response

(The authors gave the same response as above.)

Reviewer 3 Report
In the authors' response to my previous comments #1, 2, and 3, I still don't see how they resolved the issues despite that they thought they did. They should explain how they resolved them!
They should address my concern in comment #1 in the discussion section. Parts belonging to Group A and B are still not combined and compared together. The references are still not updated to 2023. The changes should be pointed out by highlighting them.
Author Response
We hope that the changes made are in accordance with your demands. Thank you for your guidance and understandingRound 4
Reviewer 1 Report
No more comments are needed.
Author Response
Thank you!